# Heart–Lung–Muscle Anti-SAE Syndrome: An Atypical Severe Combination

**DOI:** 10.3390/jcm8010020

**Published:** 2018-12-23

**Authors:** Elisabet Zamora, Elena Seder-Colomina, Susana Holgado, Bibiana Quirant-Sanchez, José Luis Mate, Eva M. Martínez-Cáceres, Ivette Casafont, Antoni Bayés-Genís

**Affiliations:** 1Cardiology Department, Hospital Universitari Germans Trias i Pujol, 08916 Badalona, Spain; elena.seder@gmail.com (E.S.-C.); abayesgenis@gmail.com (A.B.-G.); 2Departament de Medicina, Universitat Autònoma de Barcelona, 08193 Bellaterra, Spain; 3CIBERCV, Instituto de Salud Carlos III, 28029 Madrid, Spain; 4Rheumatology Department, Hospital Universitari Germans Trias i Pujol, 08916 Badalona, Spain; susholgado@gmail.com (S.H.); ivette.casafont@gmail.com (I.C.); 5Immunology Department, Hospital Universitari Germans Trias i Pujol, 08916 Badalona, Spain; bquirant@gmail.com (B.Q.-S.); evmcaceres@gmail.com (E.M.M.-C.); 6Department of Cell Biology, Physiology and Immunology, Universitat Autònoma de Barcelona, 08193 Bellaterra, Barcelona, Spain; 7Pathology Department, Hospital Universitari Germans Trias i Pujol, 08916 Badalona, Spain; jlmate.germanstrias@gencat.cat

**Keywords:** myocarditis, idiopathic inflammatory myopathy, anti-SAE antibody

## Abstract

A 78-year-old man with 3 months of progressive dyspnea, dysphony, dysgeusia, and proximal muscle weakness was diagnosed of probably idiopathic inflammatory myopathy with nonspecific interstitial pneumonia. Variable degrees of atrioventricular block and persistently elevated cardiac enzymes indicated a diagnosis of myocarditis, confirmed with cardiac magnetic resonance imaging and endomyocardial biopsy. A comprehensive immune work-up revealed anti-small ubiquitin-like modifier-1 activating enzyme (anti-SAE) antibody, a novel myositis-specific antibody, previously described mainly with overt cutaneous dermatomyositis and late skeletal muscle manifestations. Here, heart–lung–muscle involvement combined with anti-SAE antibodies was a severe combination.

## 1. Case Report

A 78-year-old man with a previous history of hypertension presented with progressive dyspnea, unexpected weight loss, Raynaud phenomenon, muscle weakness, dysphony, dysgeusia, and right hemifacial hypoesthesia, which had persisted for 3 months. A physical examination revealed bi-basal fine crackles and bilateral proximal muscle weakness in the upper and lower extremities. A dermatological examination revealed no cutaneous abnormalities. An electrocardiogram showed low voltages, sinus rhythm, first degree atrioventricular block, and QS-wave morphology in the anterior precordial leads. Bilateral interstitial infiltrates were found in the chest X-ray, and biochemical tests showed elevations in C-reactive protein (76 mg/L) and skeletal muscle and cardiac enzymes (creatine kinase: 1942 U/L, creatine kinase–muscle/brain: 50 ng/mL, and hs-troponin I: 31,125 pg/mL). The patient was admitted. An electromyogram showed signs of chronic radiculopathy L4–L5–S1 without acute axonal damage and primary affectation of muscular fiber in inferior extremities (fibrillation and positive waves in right psoas). During voluntary contraction, we observed many small polyphase complexes with an early recruitment pattern. The vastus lateralis and medial right gastrocnemius showed a big polyphasic complex with a reduced pattern without spontaneous activity. The deltoid and right biceps muscles had an interferential pattern without spontaneous activity. Capillaroscopy showed a reduced number of capillaries and avascular areas, and central nervous system magnetic resonance imaging (MRI) results were unremarkable. Thoracoabdominal computed tomography (CT) revealed nonspecific interstitial pneumonia and whole-body positron-emission tomography/CT revealed diffuse myocardial uptake. A cardiac MRI revealed mild systolic biventricular dysfunction, inferoseptal hypokinesia, biatrial dilatation, diffuse edema, and fibrosis in the atrial walls and right ventricle. These features fulfilled the Lake Louise criteria for myocarditis (Figure 1).

Coronary CT angiography ruled out coronary artery disease. An endomyocardial biopsy (EMB) was performed and showed Dallas criteria for myocarditis, with lymphocytic myocardial infiltration and moderate fibrosis (Figure 2A–C); microbiological tests were negative.

A comprehensive immunology study revealed a high titer of antinuclear antibodies (1/640), and indirect immunofluorescence showed a nucleoplasm with a speckled pattern. Antinuclear antibodies (ANA) were detected by indirect immunofluorescence (IFL) using ready-made slides from commercial sources of fixed HEp-2 cells (Nova Lite range of reagents, Inova Diagnostics). The study of autoantibodies related to the ANA pattern found was performed using a monospecific assay by inmunoblot (Euroline range of reagents, Euroimmun) in the patient’s serum being negative for Ro60, Ro52, La, RNP, and Sm. Nevertheless, given a high suspicion of autoimmune disease, the immunological study was extended to autoantibodies associated with scleroderma, such as: Scl70, PM-Scl75, PMScl100, Th/To, RNA pol III, CENP B, CENP A, fibrillarin, and Ku, using a monospecific assay by immunoblot (Euroline range of reagents, Euroimmun); being all negative. For the association of interstitial lung disease (ILD) and positive ANA pattern, we analyzed myositis-specific autoantibodies (MSA) specificities: Mi-2, MDA-5, TIF1-gamma, NXP and SAE-1, and SAE-2, the latter being positive for SAE1/SAE2, suggestive of a systemic autoimmune disease, probably inflammatory myopathy (IMM). Anti-SAE antibodies were determined using a monospecific assay by dot blot for the detection in human sera of IgG autoantibodies (Myositis12 SAE IgG, D-Tek).

Human leukocyte antigen (HLA) genotyping revealed the presence of the *HLA-DQB1*03:02* allele (measured with a real-time polymerase chain reaction assay [GenVitSet, BDR]).

Treatment was initiated with pulse-dose corticosteroid therapy for 3 days, followed by a second line immunosuppressive therapy with intravenous cyclophosphamide (900 mg), which failed to slow disease progression. Indeed, cardiac involvement symptoms prevailed in the form of variable types of arrhythmia, including atypical atrial flutter, alternating with atrial fibrillation, and second- and third-degree atrioventricular block, with an intermittent left bundle branch block; moreover, cardiac enzymes remained persistently high. Third- and fourth-line therapies with immunoglobulins and rituximab (500mg) were started, respectively. On day 28 after admission, the patient evolved with progressive breathlessness, due to progression of the interstitial infiltrate and heart failure. He required intubation but remained hypoxemic, despite mechanical ventilation, and died on day 30. The necropsy pointed to acute respiratory distress syndrome (ARDS) as the primary cause of death. It also revealed vast pulmonary fibrosis, consistent with an evolved, nonspecific interstitial pneumonia and areas of diffuse alveolar damage, with edema and hemorrhage related to ARDS (Figure 3), confirming the diagnosis of rapidly progressive interstitial lung disease (RP-ILD). The authorized thoracic limited necropsy revealed the indemnity of the intercostal muscular tissue. Cardiac histology showed diffuse myocarditis and large fibrotic areas.

## 2. Discussion

Idiopathic inflammatory myopathy is a heterogeneous group of immune mediated disorders. They are defined as chronic inflammatory muscle diseases characterized by muscle weakness and frequent extramuscular involvement [1]. The diagnosis of the different subtypes of IIM is given by the combination of clinical history, pattern of muscle involvement, and immunological profile.

Over the last decade, several myositis-specific antibodies have been described, including anti-SAE antibodies. The small ubiquitin-like modifier is a small protein, structurally similar to ubiquitin, involved in post-translational protein modifications. It is involved in several cell processes, including gene transcription and genome stability [2]. Anti-SAE antibodies were first described by Betteridge et al. [3] in 2007 and since then, they have been identified in some IIM cohorts [4]. In one of those cohorts [5], a human leukocyte antigen (HLA) profile showed that all patients with anti-SAE antibodies had at least one copy of *HLA-DQB1*03*. The patient presented here also carried the *HLA-DQB1*03* gene.

To date, anti-SAE antibodies have only been associated with dermatomyositis (DM), with a frequency range of 1%–10% [4]. Although data are scarce and heterogeneous, patients with anti-SAE antibodies have a clinical phenotype defined by prominent and often severe dermatological lesions, mild muscular symptoms, and infrequently, interstitial lung disease (ILD) and dysphagia [4]. Extramuscular symptoms have consistently been reported to be mild and nonfatal; thus, the anti-SAE syndrome was considered to have a benign course and prognosis. To our knowledge, the present study was the first to describe a case of anti-SAE probable IIM without dermatological manifestations, but with fatal RP-ILD and myocarditis.

RP-ILD is more frequently associated with amyopathic dermatomyositis, an atypical form of DM without muscular symptoms [6]. Patients with RP-ILD responded poorly to even aggressive treatment and have shown a high mortality rate. A less-studied, but critical extramuscular condition is IIM-associated cardiomyopathy. In IIMs, cardiac involvement is typically subclinical, and it can include myocarditis, rhythm disturbances, and congestive heart failure. The prevalence is difficult to estimate, due to the wide range of clinical manifestations and the lack of a systematic cardiac assessment at diagnosis. When myocarditis is suspected, a cardiac MRI is currently the main diagnostic tool [7], as highlighted by the International Consensus Group on Cardiac MR in Rheumatology [8]. A small prospective study found 55% of cardiac MRI-proven myocarditis in patients with IIM at diagnosis [9]. A complementary tool, the EMB, allows tissue microbiological testing and characterization of the inflammatory response [10]. A recent literature review found myocarditis in 38% of IIM pathology reports [11]. The clinical course and severity of IIM-related myocarditis are not well established. Our case is an example of how early, mild cardiac manifestations may lead to diffuse, persistent overt myocarditis (including arrhythmias and heart failure). To our knowledge, no previous reports have described anti-SAE antibodies associated with cardiac involvement.

Indeed, the fatal outcome we observed has raised concern regarding the risk of this IIM subset, which was previously considered benign. Taking these data into account, we suggest including heart and lung work-ups in all patients with a diagnosis of IIM, regardless of the immunologic profile.

## 3. Conclusions

To date, the presence of the anti-SAE antibody has been mainly reported as an indication of DM with mild extramuscular involvement. Here, we described a case of an anti-SAE unrelated to DM, with fatal RP-ILD, and biopsy-proven myocarditis. Further research is needed to fully understand the anti-SAE syndrome and its unpredictable outcome. This clinical case report fulfills the Spanish Protection of Personal Data Law (15/1999). Witnessed oral consent by the patient and their family were obtained.

## Figures and Tables

**Figure 1 jcm-08-00020-f001:**
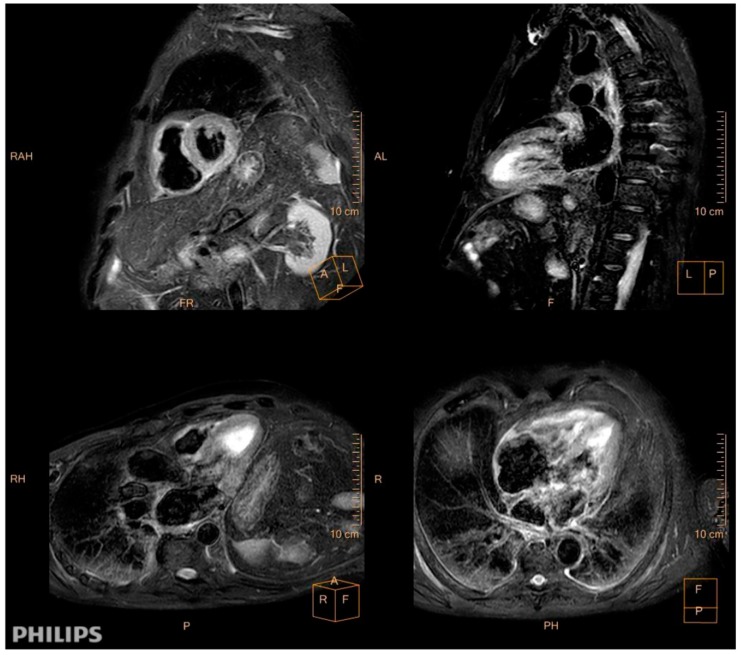
Cardiac magnetic resonance imaging (MRI). Diffuse myocardial edema in T2-weighted edema images (SA, short axis; LA, long axis; 3C, three chamber; 4C, four chamber).

**Figure 2 jcm-08-00020-f002:**
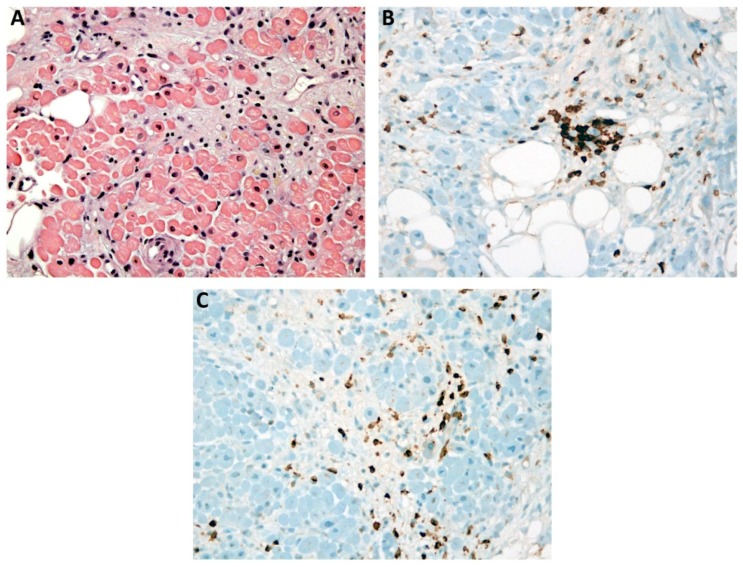
Endomyocardial biopsy. (**A**) Moderate fibrosis. Hematoxylin and eosin staining. (**B**) Lymphocyte myocardial infiltration. Immunohistochemical staining shows positive staining for CD3 (brown), which confirms the T-cell phenotype. (**C**) Macrophage myocardial infiltration. Immunohistochemical staining shows positive staining for CD68 (brown).

**Figure 3 jcm-08-00020-f003:**
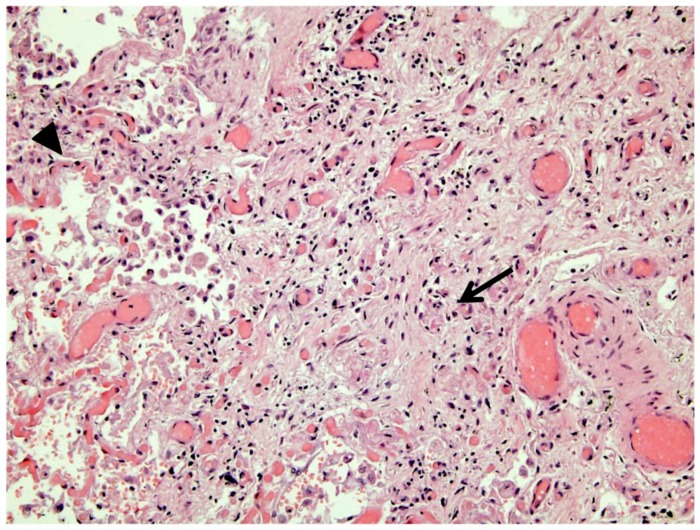
Necropsy. Pulmonary tissue. Nonspecific interstitial pneumonia consistent with evolved interstitial fibrosis (arrow) combined with areas of diffuse alveolar damage (arrowhead). Hematoxylin and eosin staining.

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
