# Peer review of "Heart–Lung–Muscle Anti-SAE Syndrome: An Atypical Severe Combination"

_jcm, 2018, doi:10.3390/jcm8010020_

Reviewer 1 Report

Thank you for allowing me to review this interesting case report. The authors report the case of 78 year old man diagnosed with idiopathic inflammatory myopathy who was positive to anti-SAE autoantibody and who developed myocarditis, interstitial pneumonitis, and finally died.

The case is interesting but there are some issues that need to be addressed by the authors.

- A muscle biopsy is mandatory in this case. We do not know if the muscle pathology findings suggest dermatomyositis, polymyositis, necrotizing myopathy or antisynthetase syndrome. Given the fact that there is ILD at necropsy this data is relevant.

- Please define the findings of the electromyography.

- The authors have to report the methods used for detection of the antibody (i.e. ELISA, blot, IP...). It would be of interest to know if other specific autoantibodies were analyzed (i.e. anti-MDA5, antisynthetase, anti-SRP...) please report the data

- This is a single case, therefore, in opinion of this reviewer statements such as included in the discussion or the abstract conclusions referring to a "reconsideration of the syndrome" should be modify or deleted.

Moreover, given that is a single case, and with a different phenotype that the usually associated with the anti-SAE it is important that all the data missing will be included

Author Response

Thank you for allowing me to review this interesting case report. The authors report the case of 78 year old man diagnosed with idiopathic inflammatory myopathy who was positive to anti-SAE autoantibody and who developed myocarditis, interstitial pneumonitis, and finally died.

The case is interesting but there are some issues that need to be addressed by the authors.

- A muscle biopsy is mandatory in this case. We do not know if the muscle pathology findings suggest dermatomyositis, polymyositis, necrotizing myopathy or antisynthetase syndrome. Given the fact that there is ILD at necropsy this data is relevant. 

- We appreciate your comment. Unfortunately, we could not make patient muscular biopsy before his death. However, we analyzed a tissue sample of intercostal muscle in authorized thoracic limited clinical necropsy and found normal muscle tissue. In the revised version of the manuscript, we added this data; Page 4, line 96: "The authorized thoracic limited necropsy revealed indemnity of the intercostal muscle tissue".

- Please define the findings of the electromyography. 

- In the revised version of the manuscript we added: "An electromyogram showed signs of chronic radiculopathy L4-L5-S1 without acute axonal damage and primary affectation of muscular fiber in inferior extremities (fibrillation and positive waves in right psoas). During voluntary contraction we observed many small polyphase complexes with early recruitment pattern. Vastus lateralis and medial right gastrocnemius showed big polyphasic complex with reduced pattern without espontaneus activity. Deltoid and right biceps muscles had interferential pattern without spontaneus activity." Page 1, line 39.

- The authors have to report the methods used for detection of the antibody (i.e. ELISA, blot, IP...). It would be of interest to know if other specific autoantibodies were analyzed (i.e. anti-MDA5, antisynthetase, anti-SRP...) please report the data

- In the revised version of the manuscript we added all this data, according to the reviewers'  suggestion. Page 3, line 64.

"Antinuclear antibodies (ANA) were detected by indirect immunofluorescence (IFL) using ready-made slides from commercial sources of fixed HEp-2 cells (Nova Lite range of reagents, Inova Diagnostics). The study of autoantibodies related to the ANA pattern found was performed using mono-specific assay by inmunoblot (Euroline range of reagents, Euroimmun) in the patient’s serum being negative for Ro60, Ro52, La, RNP and Sm. Nevertheless, given a high suspicion of autoimmune disease, the immunological study was extended to autoantibodies associated with scleroderma, such as: Scl70, PM-Scl75, PMScl100, Th/To, RNA pol III, CENP B, CENP A, fibrilarin and Ku, using a mono-specifc assay by immunoblot (Euroline range of reagents, Euroimmun); being all negative. For the association of interstitial lung disease (ILD) and positive ANA pattern we analyzed MSA specificities: Mi-2, MDA-5, TIF1-gamma, NXP and SAE-1 and SAE-2,the latter being positivity for SAE1/SAE2, suggestive of a systemic autoimmune disease, most likely an idiopathic inflammatory myopathy (IMM). AntiSAE antibodies were determined using mono-specific assay by dot blot for the detection in human sera of IgG autoantibodies (Myositis12 SAE IgG, D-Tek)."

 - This is a single case, therefore, in opinion of this reviewer statements such as included in the discussion or the abstract conclusions referring to a "reconsideration of the syndrome" should be modify or deleted. 

According to the reviewer's suggestion in the revised version of the manuscript we have removed this conclusion in the abstract: "The clinical phenotype and prognosis of anti-SAE syndrome might require reconsideration" Page 1, line 25. In the discussion we also removed "The unique combination of clinical findings in this case suggests an invitation to review the definition of the anti-SAE syndrome". Page 5, line 140. Finally we modified the final conclusion: we have changed "Based on this findings, the clinical phenotype and prognosis of anti-SAE syndrome might require reconsideration" by  "Further research is needed to understand fully the anti-SAE syndrome and its unpredictable outcome". Page 5, line 149-150.

- Moreover, given that is a single case, and with a different phenotype that the usually associated with the anti-SAE it is important that all the data missing will be included 

- In the revised version of the manuscript, we have included all the data required in previous reviewers' comments.

Reviewer 2 Report

The case is of interest in extending the clinical phenotype of myositis associated with anti-SAE autoantibodies - such severe manifestations of myocarditis and progressive pulmonary disease have not been previously reported. Also the accompanying imaging and histology is of interest. Some modifications would improve the manuscript.

I would avoid the word 'dreadful' in title and elsewhere. Severe may suffice

It is debatable as to whether one case should lead to reconsidering a syndrome - but watching for atypical cases is justified - may be some minor rewording required

The authors describe rapidly progressive lung disease in the discussion (RPLD). If the authors believe the case had RPLD it would be helpful to include this in the case report itself.

The assay used to detect anti-SAE needs to be mentioned. RPLD in myositis is usually associated with anti-MDA5 - presumably anti-MDA5 was tested for as well

Author Response

The case is of interest in extending the clinical phenotype of myositis associated with anti-SAE autoantibodies - such severe manifestations of myocarditis and progressive pulmonary disease have not been previously reported. Also the accompanying imaging and histology is of interest. Some modifications would improve the manuscript.

- I would avoid the word 'dreadful' in title and elsewhere. Severe may suffice

- According to the reviewer's suggestion in the revised version of the manuscript we have changed "dreadful" by  "severe" in the title and in the abstract (page 1, line 24). 

- It is debatable as to whether one case should lead to reconsidering a syndrome - but watching for atypical cases is justified - may be some minor rewording required

- According to the reviewer's suggestion in the revised version of the manuscript we have removed this conclusion in the abstract: "The clinical phenotype and prognosis of anti-SAE syndrome might require reconsideration" Page 1, line 25. In the discussion we also removed "The unique combination of clinical findings in this case suggests an invitation to review the definition of the anti-SAE syndrome". Page 5, line 140. Finally we modified the final conclusion: we have changed "Based on this findings, the clinical phenotype and prognosis of anti-SAE syndrome might require reconsideration" by  "Further research is needed to understand fully the anti-SAE syndrome and its unpredictable outcome". Page 5, line 149-150.

 - The authors describe rapidly progressive lung disease in the discussion (RPLD). If the authors believe the case had RPLD it would be helpful to include this in the case report itself.

- As suggested by the reviewer we have added in the case "...confirming the diagnosis of rapidly progressive interstitial lung disease (RP-ILD)" Page 4, Line 96.

- The assay used to detect anti-SAE needs to be mentioned. RPLD in myositis is usually associated with anti-MDA5 - presumably anti-MDA5 was tested for as well

- In the revised version of the manuscript we added all this data, according to the reviewers'  suggestion. Page 3, line 64.

"Antinuclear antibodies (ANA) were detected by indirect immunofluorescence (IFL) using ready-made slides from commercial sources of fixed HEp-2 cells (Nova Lite range of reagents, Inova Diagnostics). The study of autoantibodies related to the ANA pattern found was performed using mono-specific assay by inmunoblot (Euroline range of reagents, Euroimmun) in the patient’s serum being negative for Ro60, Ro52, La, RNP and Sm. Nevertheless, given a high suspicion of autoimmune disease, the immunological study was extended to autoantibodies associated with scleroderma, such as: Scl70, PM-Scl75, PMScl100, Th/To, RNA pol III, CENP B, CENP A, fibrilarin and Ku, using a mono-specifc assay by immunoblot (Euroline range of reagents, Euroimmun); being all negative. For the association of interstitial lung disease (ILD) and positive ANA pattern we analyzed MSA specificities: Mi-2, MDA-5, TIF1-gamma, NXP and SAE-1 and SAE-2,the latter being positivity for SAE1/SAE2, suggestive of a systemic autoimmune disease, most likely an idiopathic inflammatory myopathy (IMM). AntiSAE antibodies were determined using mono-specific assay by dot blot for the detection in human sera of IgG autoantibodies (Myositis12 SAE IgG, D-Tek)." 

Round  2

Reviewer 1 Report

The article has been improved.

Minor changes: in absence of confirmatory muscle biopsy I suggest to state "probable inflammatory myopathy" instead of "inflammatory myopathy", or alternatively "systemic disease with associated muscle inflammation

Author Response

According to the reviewer's suggestion in the revised version of the manuscript we have changed "inflammatory myopathy" by "probable inflammatory myopathy". Page 1 line 18, Page 3 Line 75 and Page 5 line 122.